

**Variability of vertical structure of precipitation with sea surface temperature over the**
**Arabian Sea and the Bay of Bengal as inferred by TRMM PR measurements**
**Kadiri Saikranthi[1], Basivi Radhakrishna[2], Thota Narayana Rao[2] and**
**Sreedharan Krishnakumari Satheesh[3]**
[1] *Department of Earth and Climate Science, Indian Institute of Science Education and*
*Research (IISER), Tirupati, India.*
[2] *National Atmospheric Research Laboratory, Department of Space, Govt. of India, Gadanki*
*- 517112, India.*
[3] *Divecha Centre for Climate Change, Centre for Atmospheric and Oceanic Sciences, Indian*
*Institute of Science, Bangalore - 560012, India.*

**Address of the corresponding author**
Dr. K. Saikranthi,
Department of Earth and Climate Science,
Indian Institute of Science Education and Research (IISER),
Tirupati,
Andhra Pradesh, India.
Email: ksaikranthi@gmail.com



**Abstract**

Tropical Rainfall Measuring Mission (TRMM) Precipitation Radar (PR) 2A25 reflectivity profiles data during the period 1998 - 2013 are used to study the differences in the vertical structure of precipitation and its variation with sea surface temperature (SST) over the Arabian Sea (AS) and the Bay of Bengal (BOB). Even though the AS and the BOB are parts of the Indian Ocean, they exhibit distinct features in vertical structure of precipitation and its variation with SST. The variation of reflectivity and precipitation echo top occurrence with SST is remarkable over the AS but trivial over the BOB. The median reflectivity increases with SST at all heights below 10 km altitude, but the increase is prominent below the freezing level height over the AS. On the other hand, irrespective of altitude, reflectivity profiles are same at all SSTs over the BOB. To understand these differences, variation of aerosols, cloud and water vapor with SST is studied over these seas. At SSTs less than 27°C, the observed high aerosol optical depth (AOD) and low total column water vapor (TCWV) over the AS results in small Cloud effective radius (CER) values and low reflectivity. As SST increases AOD decreases and TCWV increases, which result in large CER and high reflectivity. Over the BOB the change in AOD, TCWV and CER with SST is marginal. Thus, the observed variations in reflectivity profiles seem to be present from the cloud formation stage itself over both the seas.



## 1. Introduction


Indian summer monsoon (ISM) is one of the most complex weather phenomena,
involving coupling between the atmosphere, land and ocean. At the boundary of the ocean
and atmosphere air-sea interactions play a key role for the coupled Earth system (Wu and
Kirtman 2005; Feng et al. 2018). SST – precipitation relations are the important measures for
the air-sea interactions on different temporal scales (Woolnough et al., 2000; Rajendran et al.
2012). Recent studies (Wang et al. 2005; Rajeevan et al. 2012; Chaudhari et al. 2013;
Chaudhari et al. 2016) have shown that the simulation of ISM can be improved with the exact
representation of sea surface temperature (SST) - precipitation relationship.
The dynamics of Madden-Julian oscillation (MJO) campaign (DYNAMO) portrayed
the importance of understanding the link between SST and convective initiation at MJO
scales (Yoneyama et al. 2013). With known differences in SST between Western Pacific and
Indian Ocean, Barnes and Houze (2013) showed the occurrence of shallow systems
maximized during the suppressed phases of MJO while the deep wide convective systems
occurred during the active phases of MJO. SST modulates the meteorological factors that
influence the formation and evolution of different kinds of precipitating systems over tropical
oceans (Gadgil et al. 1984; Schumacher and Houze, 2003; Oueslati and Bellon 2015).
The relationships between the SST and cloud/precipitation have been studied in
variety of contexts during the past three decades. The non-linear relationship of SST-
precipiation/cloud occurrence (Gadgil et al. 1984; Woolnough et al., 2000; Rajendran et al.
2012; Sabin et al. 2012; Meenu et al. 2012; Nair and Rajeev 2014; Roxy 2014; Nair et al.
2017) is well documented over the Indian Ocean. The probability of organized convection
increases with SST up to a critical value of ~ 28°C (Gadgil et al. 1984). Sabin et al. (2012)
and Meenu et al. (2012) showed that the convection is no longer dependent on SST at SSTs
greater than 30°C. Later, by considering the time lag between the SST and rainfall Roxy



(2014) argued that this upper threshold can exceed till 31°C. Sengupta et al. (2001) showed
that the intraseasonal variability of SST is not same over the entire Indian Ocean. Later, Roxy
et al. (2013) estimated the time lag for SST and precipitation to be 2 and 5 weeks for the Bay
of Bengal (BOB) and the Arabian Sea (AS), respectively. Through this study they found that
the response of precipitation to SST anomalies is faster over the AS than the BOB. Also, the
summer monsoon experiment (MONEX) showed the influence of the AS and the BOB on the
rainfall produced over the Indian sub-continent (Krishnamurti 1985; Houze and Churchill
1987) and also proved how these two seas are different with respect to the other oceans in
terms of SST, back ground atmosphere and the occurrence of precipitating systems.

Knowing the differences in atmospheric conditions over the AS and the BOB during

June and September (JJAS) the occurrence of various kinds of precipitating systems over
these two seas is studied in Liu et al. (2007), Romatschke et al. (2010), Saikranthi et al.
(2014), Houze et al. (2015). These studies showed that the occurrence of shallow systems is
prevalent over the Arabian Sea while deeper systems are abundant in the Bay of Bengal.
Recently, Saikranthi et al. (2018) showed that the observed differences in the occurrence of
various kinds of precipitating systems is exist even in El Niño and La Niña periods also, but
with variable magnitudes. Aforementioned studies mainly focussed on the variation of
surface rainfall, morphology of vertical structure of precipitation, occurrence of cloudiness
with SST over the Indian Ocean. But, none of them studied the variation of vertical structure
of precipitation (in terms of occurrence and intensity) with SST. The strength of the
convective forcing strongly depends on SST (Sabin et al. 2012) and changes in SST have the
potential of altering the vertical structure of precipitation (Oueslati and Bellon 2015).

The vertical structure of precipitation information is essential for improving the

accuracy of rainfall estimation (Fu and Liu 2001; Sunilkumar et al. 2015), understanding the
dynamical and microphysical processes of hydrometeor growth/decay mechanisms (Houze



2004; Greets and Dejene 2005; Saikranthi et al. 2014; Rao et al. 2016) and flash rates (Liu et
al. 2012), and for improving the latent heating retrievals (Tao et al. 2006). Also, most of the
earlier studies dealing with SST and cloud/precipitation population considered whole Indian
ocean as a single entity. But in reality the BOB and the AS of Indian ocean possesses
distinctly different features, like SST and its variability over seasonal and intraseasonal scales
(Sengupta et al. 2001; Roxy et al. 2013), the monsoonal wind speeds (Findlater 1969) and
also the type of rain (Liu et al. 2007; Romatschke et al. 2010; Saikranthi et al. 2014; Rao et
al. 2016). Knowing the importance of vertical structure of precipitation and SST modulation
of background atmospheric conditions, in the present study we have studied the variation of
vertical structure of precipitation with SST and their causative mechanisms over different
regions of Indian ocean, in particular over the BOB and the AS.

The present paper is organized as follows. Section 2 describes the data and method of

analysis. The variation of the vertical structure of precipitation with SST over BOB and AS is
studied in section 3. Section 4 discusses the factors influencing the variation of vertical
structure over BOB and AS. The results are summarized in Section 5.
**2. Data**

The present study utilizes 16 years (1998-2013) of Tropical rainfall measuring

mission (TRMM) precipitation radar (PR) 2A25 (version 7) dataset during the southwest
monsoon season (June to September). TRMM-PR dataset comprising vertical profiles of
attenuation corrected reflectivity with 17 dBZ as minimum detectable signal (Iguchi et al.
2009). Comparing TRMM-PR data with Kwajalein S-band radar data Schumacher and Houze
(2000) showed that TRMM-PR misses 15% of the echo area observed above 0°C levels due
to the sensitivity threshold (17 dBZ). Through this study they concluded that TRMM-PR
highly under samples weaker echoes from ice particles associated with stratiform rain aloft
but manages to capture most of the near-surface precipitation accumulation. The range



resolution of TRMM-PR reflectivity profiles is 250 m with a horizontal footprint size of ~4.3
and 5 km before and after the boosting of its orbit, respectively. It scans ±17° from nadir with
a beam width of 0.71° covering a swath of 215 km (245 km after the boost). TRMM-PR data
uniqueness is its ability in pigeonholing the precipitating systems into convective, stratiform
and shallow rain. This classification is based on two methods namely the horizontal method
(H - method) and the vertical method (V - method) using the bright band identification and
the reflectivity profile (Awaka et al. 2009). The original TRMM-PR 2A25 vertical profiles of
attenuation corrected reflectivity are gridded to a three dimensional Cartesian coordinate
system with a spatial resolution of $0.05° \times 0.05°$. The detailed methodology of interpolating
the TRMM-PR reflectivity data into the 3D Cartesian grid is discussed in Houze et al. (2007).
This     dataset     is     available     at     the     University     of     Washington     website
(http://trmm.atmos.washington.edu/).

To understand the observed variations in the vertical structure of precipitation in the

light of microphysics of clouds, Moderate Resolution Imaging Spectroradiometer (MODIS)
AQUA satellite level 3 data (MYD08) are considered. In particular, the daily atmospheric
products of aerosol optical depth (AOD) (Hubanks et al. 2008), cloud effective radius (CER)
ice, and CER liquid (Platnick et al. 2017) during the period 2003 and 2013 have been used.
MODIS AOD dataset is a collection of aerosol optical properties at 550 nm wavelength, as
well as particle size information. Level 2 MODIS AOD is derived from radiances using either
one of the three different algorithms, i.e., over ocean Remer et al. (2005) algorithm, over land
the Dark-Target (Levy et al. 2007) algorithm and for brighter land surfaces the Deep-Blue
(Hsu et al. 2004) algorithm. CER is nothing but the weighted mean of the size distribution of
cloud drops i.e., the ratio of third moment to second moment of the drop size distribution. In
the level 3 MODIS daily dataset, aerosol and cloud products of level 2 data pixels with valid
retrievals within a calendar day are first aggregated and gridded to a daily average with a



spatial resolution of 1° × 1°. For CER grid box values, CER values are weighted by the
respective ice/liquid water cloud pixel counts for the spatiotemporal aggregation and
averaging processes.
The background atmospheric structure and SST information are taken from the
European Centre for Medium Range Weather Forecasting (ECMWF) Interim Reanalysis
(ERA). ERA-Interim runs 4DVAR assimilation twice daily (00 and 12 UTC) to determine the
most likely state of the atmosphere at a given time (analysis). The consistency across
variables in space and in time (during 12-hour intervals) is thus ensured by the atmospheric
model and its error characteristics as specified in the assimilation. ERA-Interim is produced
at T255 spectral resolution (about 0.75°, ~ 83 km) with a temporal resolution of 6h for upper
air fields and 3h for surface fields. The performance of the data assimilation system and the
strengths and limitations of ERA-Interim datasets are found in Dee et al. (2011). The original
0.75° × 0.75° spatial resolution gridded dataset is rescaled to a resolution of 0.125° × 0.125°.
The temporal resolution of the dataset used is 6h (00, 06, 12 and 18 UTC).
The variation of vertical structure of precipitation with SST are studied by considering
the dataset between 63°E - 72°E and 8°N-20°N over the AS and 83°E - 92°E and 8°N - 21°N
over the BOB. These regions of interest along with the SWM seasonal mean SST over the
two seas are depicted in Fig. 1. These regions are selected in such a way that the costal
influence on SST is eluded from the analysis. As small amount of rainfall is observed over
the western AS (west to 63°E latitude) during SWM (Saikranthi et al. 2018), this region is
also not considered in the present analysis. The seasonal mean SST is higher over the BOB
than in the AS by more than 1 °C during the SWM season corroborating the findings of
Shenoi et al. (2002). The nearest space and time matched SST data from ERA-Interim are
assigned to the TRMM-PR and MODIS observations for further analysis.



### 3. Variation of vertical structure of precipitation with SST


The occurrence (in terms of %) of conditional precipitation echoes (Z ≥ 17 dBZ) at
different altitudes as a function of SST over the AS and the BOB is shown in Fig. 2. The
variation of precipitation echoes occurrence frequency with SST is quite different over both
the seas. It increases with increase in SST over the AS, but remains nearly same over the
BOB. Higher occurrence of precipitation extends to higher heights with increasing SST over
the AS, while such variation is not quite evident over the BOB. Precipitation echoes are
confined to 8 km at lower SST (< 28° C) over the AS, but exhibits a gradual rise in height
with increase in SST. Confinement of echoes to lower heights at lower SST is mainly due to
the abundant occurrence of shallow systems over the AS (Saikranthi et al. 2014; Rao et al.
2016). Interestingly, high occurrence of precipitation is seen at higher heights even at lower
SSTs over the BOB, indicating the presence of deeper storms. Such systems exist at all SST's
over the BOB.
To examine the variation of reflectivity profiles with SST, median profiles of
reflectivity in each SST bin are computed over the AS and the BOB and are depicted in Figs.
3a & 3d, respectively. The space- and time-matched conditional reflectivity profiles are
grouped into 1°C SST bins and then the median is estimated at each height, only if the
number of conditional reflectivity pixels (Figs. 3c & 3f) is greater than 500. It is clear from
Figs. 3a & 3d that the median reflectivity profiles are distinctly different over the AS and the
BOB, even at same SST. Over the AS, reflectivity profiles show only small variations (≤ 1
dBZ) with SST above (> 5 km) the melting region, but vary significantly below the melting
level (< 5 km). These variations in reflectivity profiles with SST are negligible over the BOB.
Below the melting layer, the reflectivity increases from 24 dBZ to ~28 dBZ with increase in
SST from 26°C to 30°C over the AS, but it is almost the same (~28dBZ) at all SST's over
the BOB. The standard deviation of reflectivity also exhibits similar variation as that of



median profiles with SST over the AS and the BOB. In general the standard deviation of
reflectivity, representing the variability in reflectivity within the SST bin, is larger over the
BOB than the AS.
The median reflectivity profiles show a gradual increase with decreasing altitude from
~ 10 km to 6 km and an abrupt enhancement is seen just below 6 km over both the seas. The
sudden enhancement at the freezing level is primarily due to the aggregation of hydrometeors
and change in dielectric factor from ice to water (Fabry and Zawadzki 1995; Rao et al. 2008;
Cao et al. 2013). Below the bright band, the raindrops can grow by collision-coalescence
process and reduce their size either by breakup or by evaporation processes. The collision -
coalescence results in negative slope in the reflectivity profile, whereas breakup and
evaporation results in positive slope (Liu and Zipser 2013; Cao et al. 2013; Saikranthi et al.
2014). The observed negative slope in the median reflectivity profiles below the bright band
region indicates the low-level hydrometeor growth over both the seas. This hydrometeor
growth below melting region indicates the predominance of collision-coalescence process
than the collision-breakup process over both the seas. The magnitude of the slope is nearly
equal over both the seas, indicating that the rate of growth, on average, is nearly equal.
**4. Factors affecting the vertical variation of reflectivity with SST**
The formation and evolution of precipitating systems depends on the stability of the
boundary layer, dynamics and thermodynamics of the ambient atmosphere. To know the
stability of the marine boundary layer at various SSTs the lower tropospheric stability (LTS)
is considered. LTS is defined as the difference in potential temperature between 700 hPa
($\theta_{700}$) and surface ($\theta_0$) i.e., $LTS = \theta_{700} - \theta_0$ that represents the strength of the inversion caps by
the planetary boundary layer (Wood and Bretherton 2006). The LTS values were computed
from the ERA-Interim temperature data during SWM season over the selected regions and
are depicted in Fig. 4(a). LTS decreases with SST up to 29°C and increases a little at further



SSTs over both the seas however when compared to the BOB the LTS values are larger over
the AS at all SSTs. The stability of the planetary boundary layer is very high at lower SSTs
and as SST increases the stability decreases drastically over the AS up to 29°C and increases
a little at further SSTs. On the other hand the variability in planetary boundary layer stability
with SST is trivial over the BOB. Also shown in Fig. 4(b) is the convective available
potential energy (CAPE) at different SSTs over both the Seas. CAPE is calculated following
Emanuel (1994). CAPE increases with rise in SST over both the seas while its magnitude is
relatively large over the BOB than the AS at all SSTs. The large LTS and small CAPE values
at lower SSTs over the AS don't allow the precipitating systems to grow to higher altitudes
and in turn precipitate in the form of warm rain. As SST increases LTS decreases drastically
and CAPE increases and hence the precipitating systems can grow to higher altitudes.
Though LTS increases above 29°C the instability created by the large CAPE can penetrate
the planetary boundary layer and favours the formation of deeper systems. On the other hand
LTS values are lower and remain almost same at all SSTs and large CAPE values over the
BOB are conducive for the precipitating systems to grow to higher altitudes as depicted in
Fig. 2.

The observed differences in reflectivity profiles of precipitation with SST could be

originated at the cloud formation stage or in the evolution stage or due to both. In order to
understand this, the variation of mean CER for ice and liquid at different SST's over the AS
and the BOB is depicted in Figs. 5a & 5b, respectively. The mean is calculated only when the
number of data points is larger than 100 in each SST bin. It is evident from Figs. 5a & 5b that
both CER ice and liquid increase with rise in SST substantially over the AS but the increase
is marginal over the BOB. For example, as SST rises from 26°C to 31°C, the CER ice and
liquid vary from 20 μm to 32 μm and 14.7 μm to 20.8 μm, respectively over the AS, whereas
they vary, respectively, from 29 μm to 31 μm and 18.5 μm to 19.5 μm over the BOB. Also,



the cloud droplets are small in size at lower SSTs and bigger at higher SSTs over the AS,
whereas they are big over the BOB irrespective of SST. These smaller sized hydrometeors at
low SSTs are responsible for the observed small reflectivities above the melting layer over
the AS than the BOB as reflectivity is more sensitive to the particle size than the droplet
concentration ($Z \propto D^6$). At higher SSTs, the CER values are approximately equal over both
the seas and in turn the observed reflectivities (Fig. 5). This suggests that the variations seen
in vertical profiles of reflectivity are originating in the cloud itself.
Numerous studies have examined the aerosol effects on cloud formation through
heterogeneous nucleation and precipitation (Twomey 1977; Albrecht 1989; Tao et al. 2012;
and Rosenfeld et al. 2014). For fixed liquid water content, as the concentration of aerosols
increases, the number of cloud drops increases and droplet size reduces (Twomey 1977).
Utilizing the aircraft measurements over Indian sub-continent Ramanathan et al. (2001)
showed that the cloud drop number density increase with increasing aerosol number density
both over continental and maritime regions. Connolly et al. (2009), Li and Min (2010),
Niemand et al. (2012), Creamean et al. (2013), and  Fan et al. (2014) showed that dust also
act as ice nuclei through heterogeneous nucleation and these ice nuclei directly change the ice
nucleation processes that determine the initial number concentration and size distribution of
ice crystals. Thus, to understand the role of aerosols in the observed variations in the CER
with SST, the seasonal mean AOD variation with SST is plotted in Fig. 6a for the SWM.
AOD decreases from 0.62 to 0.31 with rise in SST from 26°C to 31°C over the AS but only
0.42 to 0.36  as SST varies from 27°C to 30°C and then increases with rise in SST over the
BOB. Also shown in Fig. 6b is the variation of total column water vapor (TCWV) with SST
over both the seas. TCWV shows a gradual increase with SST over the AS while it decreases
initially from 27°C to 28°C, and then increases with SST over the BOB. At a given SST the
TCWV is more in the BOB than in the AS. More number of aerosols and relatively low



TCWV over the AS results in large number of cloud drops with reduced size (Twomey 1977;
Ramanathan 2001). These reduced size cloud drops are responsible for the observed small
CER values at SSTs less than 28°C. As SST rises the AOD decreases and TCWV increases
such that the cloud particles grow in size which in turn increases CER. On the other hand, the
change in AOD and TCWV (and as a result in CER) is not prominent with SST over the
BOB, as seen in the Fig. 5.

To understand the transport of aerosols at low and mid-levels the wind magnitudes

and directions at 850 hPa and 500 hPa levels are shown in Fig. 7. The strong lower
tropospheric winds produce sea salt particles as well as transport dust from the Horn of
Africa and the mid tropospheric winds transport dust from the Arabian Desert over the AS
(Li and Ramanathan 2002). On the other hand the continental aerosols from India landmass
are transported to the BOB both at low and mid troposphere. Satheesh et al. (2006) showed
an increase in AOD with increase in latitude over the AS due to the dust advection from
Arabia desert regions during SWM season, whereas SST decreases with increase in the
latitude. In other words the SST is low and AOD is high in northern AS whereas over the
southern AS, SST is high and AOD is low. This contrasting spatial distribution of AOD and
SST could cause a negative correlation between AOD and SST. To examine whether the
observed decrease in AOD with increase in SST over the AS is due to the latitudinal variation
of AOD or exists at all latitudes, we have segregated the data into 2° latitude bins and plotted
the mean AOD with SST for all bins and is depicted in Fig. 8a. In spite of the magnitude,
AOD variation with SST is nearly similar at all latitudes of the AS, i.e., the higher AOD is
observed at lower SSTs and vice versa. On the other hand the latitudinal variation of AOD
with SST over the BOB shown in Fig. 8b also show a decrease in AOD with SST up to 30°C
but the magnitude of variation is trivial relative to the AS. As also depicted in Fig. 6a above
30°C AOD increases with SST over the BOB. This indicates that though there is a difference



in magnitude of variation, AOD varies with SST over both the seas at all latitudes. This
analysis is repeated using the multi-angle imaging spectroradiometer (MISR) dataset (which
is not shown here) for small, medium large aerosol particles. Interestingly all three types also
show a decrease in AOD with rise in SST over both the seas.
**5. Conclusions**

Sixteen years of TRMM-PR 2A25 reflectivity profiles and 11 years of MODIS AOD

and CER data are utilized to understand the differences in variation of vertical structure of
precipitation with SST over AS and BOB. This analysis reveals that the variation of
reflectivity with SST is remarkable over the AS and marginal over the BOB. The reflectivity
increases with rise in SST over the AS and remains the same at all SSTs over the BOB. This
change in reflectivity over the AS is more prominent below the freezing level height (~ 4
dBZ) than the above (~ 1 dBZ). Over the AS, the abundance of aerosols and less moisture at
SSTs < 27°C result in high concentration of small diameter cloud droplets.  As SST increases
the aerosol concentration decreases and moisture increases such that the bigger cloud droplets
are formed. Thus, the reflectivity increases with rise in SST over the AS. On the other hand,
AOD, TCWV and CER do not show substantial variation with SST over the BOB and hence
the change in reflectivity is small. Over the BOB, the mid troposphere is wet and
hydrometeor's size at the formation stage is nearly the same at all SSTs.  The evolution of
hydrometeors during their descent is also similar at all SST's, as evidenced by nearly similar
reflectivity profiles.
**Acknowledgements**
The authors would like to thank Prof. Robert Houze and his team for the interpolated 3D
gridded TRMM-PR dataset (http://trmm.atmos.washington.edu), ECMWF (http://data-
portal.ecmwf.int/) team for providing the ERA-Interim dataset and MODIS
(https://ladsweb.modaps.eosdis.nasa.gov/) science team for providing the AOD and CER



dataset. The authors express their gratitude to Prof. J. Srinivasan for his fruitful discussions
and valuable suggestions in improving the quality of the manuscript. The corresponding
author would like to thank Department of Science & Technology (DST), India for providing
the financial support through the reference number DST/INSPIRE/04/2017/001185.

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





**Figure captions**
**Figure 1:** Spatial distribution of SWM mean SST (in °C) obtained from ERA-Interim
reanalysis data over the AS and the BOB. The regions considered in this analysis over
these two seas are shown with the boxes.
**Figure 2:** (a) and (b) represent the altitudinal distribution of occurrence of conditional
reflectivity ($\geq$ 17 dBZ) as a function of SST with respect to precipitation occurrence at
that particular SST interval over the AS and the BOB, respectively.
**Figure 3:** (a), (d) and (b), (e) represent vertical profiles of median reflectivity and their
standard deviation (in dBZ) with SST over the AS (63°E-72°E & 8°N-20°N) and the
BOB (83°E-92°E & 8°N-21°N), respectively during the SWM season. (c) and (f)
show the number of conditional reflectivity pixels at each altitude used for the
estimation of the median and standard deviation.
**Figure 4: T**he variation of mean LTS with SST over the AS (63°E-72°E & 8°N-20°N) and
the BOB (83°E-92°E & 8°N-21°N) during the SWM season.
**Figure 5:** (a) and (b), respectively, represent the variation of mean CER ice (in μm) and
mean CER liquid (in μm) with SST over the AS (63°E-72°E & 8°N-20°N) and the
BOB (83°E-92°E & 8°N-21°N) during the SWM season.
**Figure 6:** (a) The variation of mean AOD and (b) TCWV (in mm) with SST over the AS
(63°E-72°E & 8°N-20°N) and the BOB (83°E-92°E & 8°N-21°N) during SWM.
**Figure 7:** Winds during the SWM season at 850 hPa and 500 hPa levels. The shading colors
represent the magnitude of the wind and arrow indicates the direction of the wind.
**Figure 8:** Latitudinal variation (for every 2° latitude interval) of mean aerosol optical depth
over Arabian Sea averaged over 63-72°E.






**Figures**

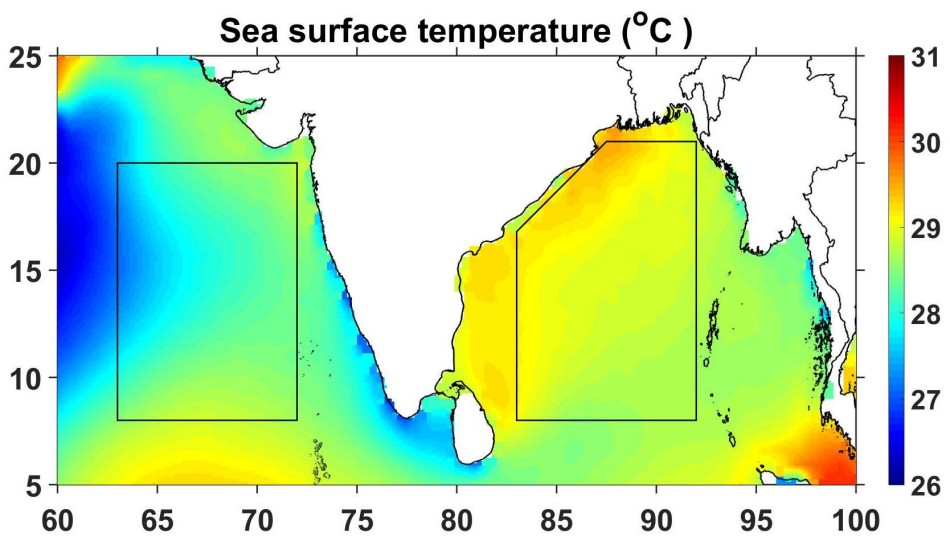


**Figure 1:** Spatial distribution of SWM mean SST (in °C) obtained from ERA-Interim
reanalysis data over the AS and the BOB. The regions considered in this analysis over
these two seas are shown with the boxes.



















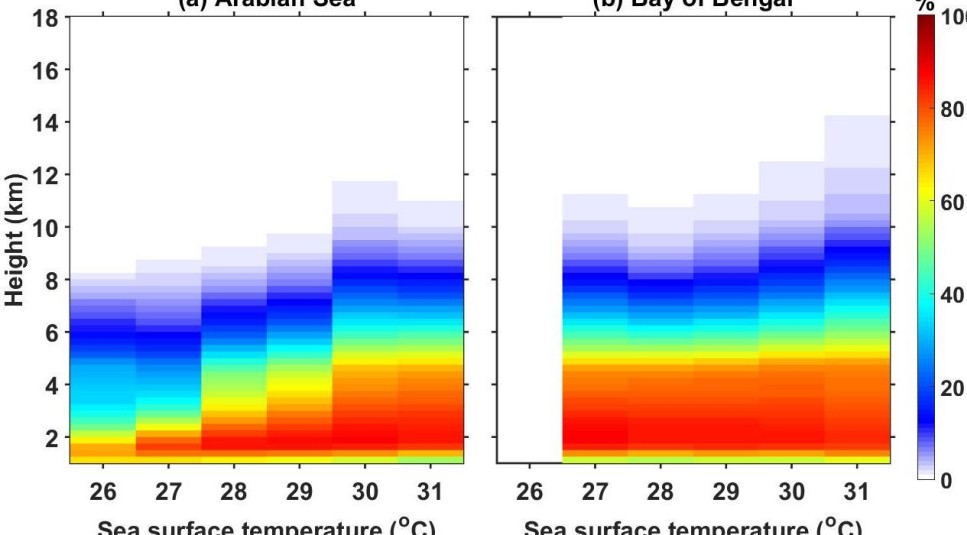



**Figure 2:** (a) and (b) represent the altitudinal distribution of occurrence of conditional
reflectivity (≥ 17 dBZ) as a function of SST with respect to precipitation occurrence at
that particular SST interval over the AS and the BOB, respectively.


















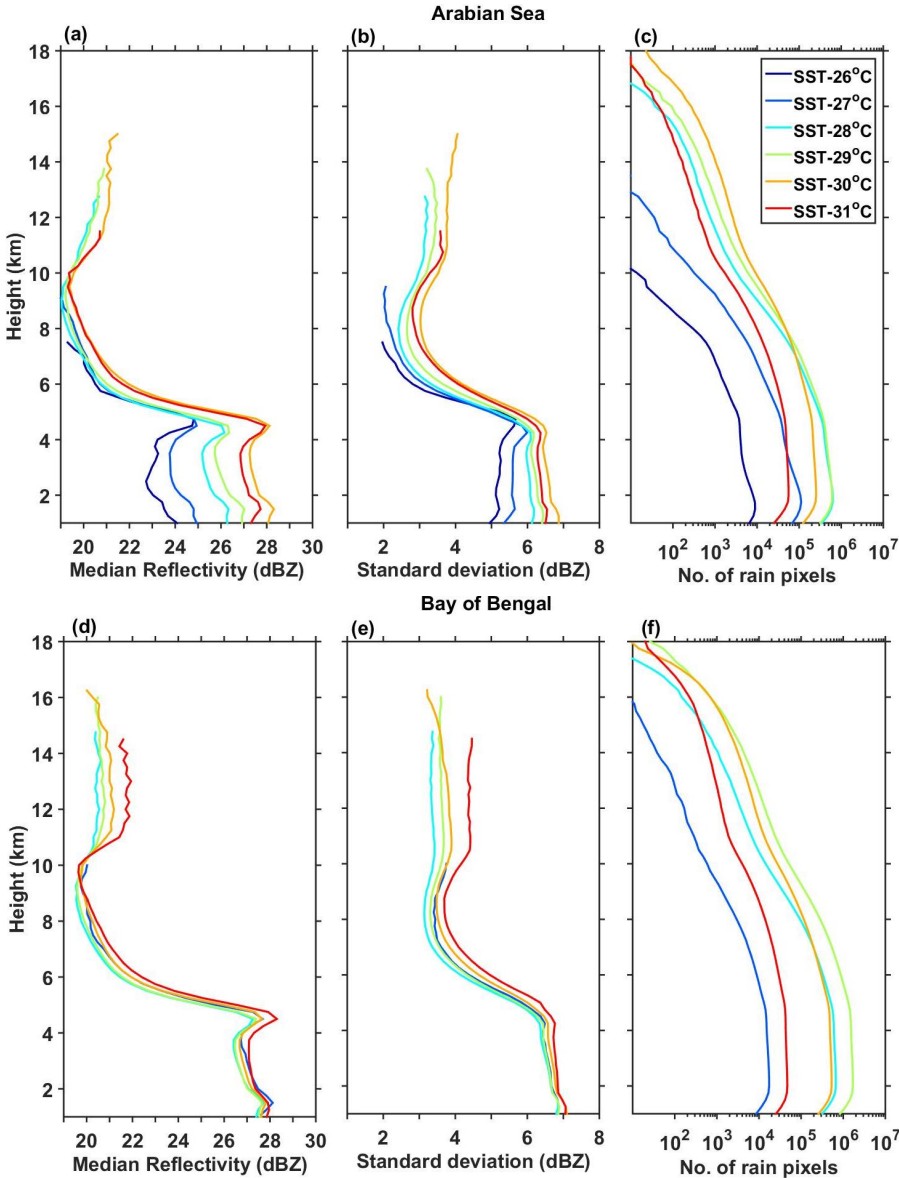

**Figure 3:** (a), (d) and (b), (e) represent vertical profiles of median reflectivity and their standard deviation (in dBZ) with SST over the AS (63°E-72°E & 8°N-20°N) and the BOB (83°E-92°E & 8°N-21°N), respectively during the SWM season. (c) and (f) show the number of conditional reflectivity pixels at each altitude used for the estimation of the median and standard deviation.






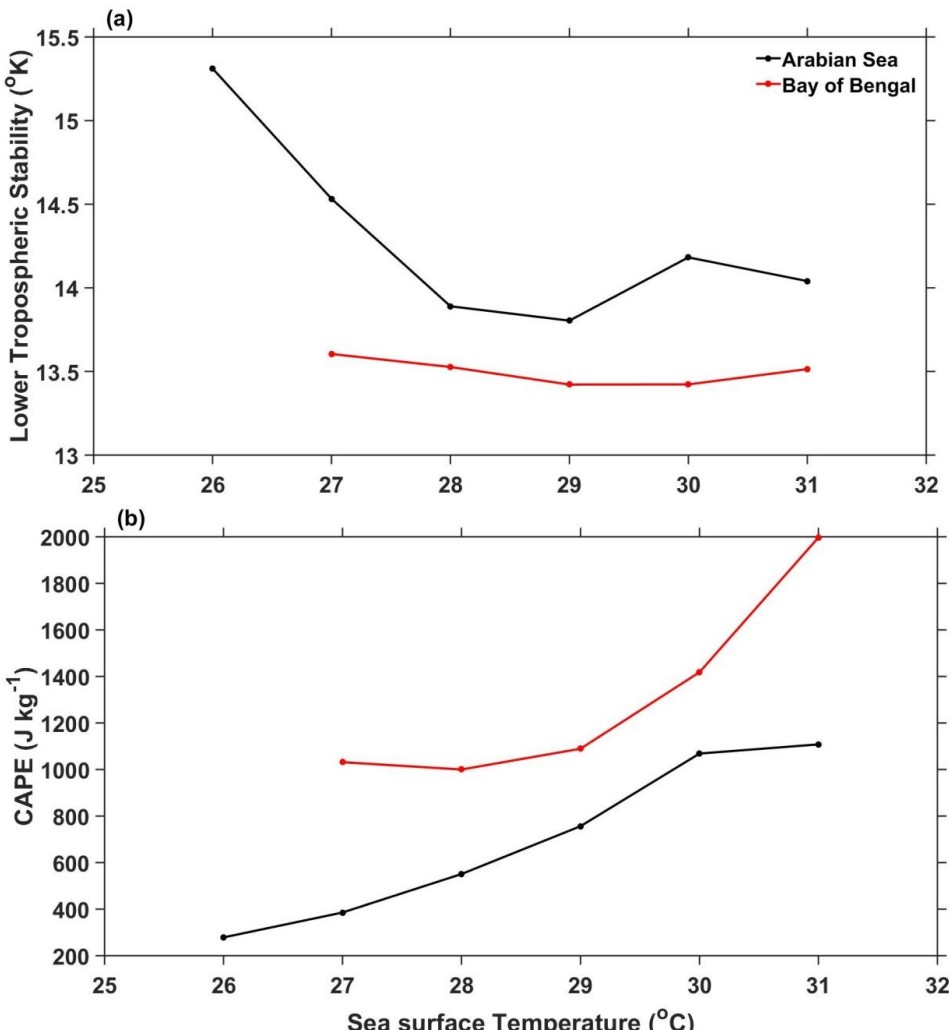



**Figure 4:** (a) The variation of mean LTS (in °K) with SST over the AS (63°E-72°E & 8°N-20°N) and the BOB (83°E-92°E & 8°N-21°N) during the SWM season. (b) Same as (a) but for CAPE.









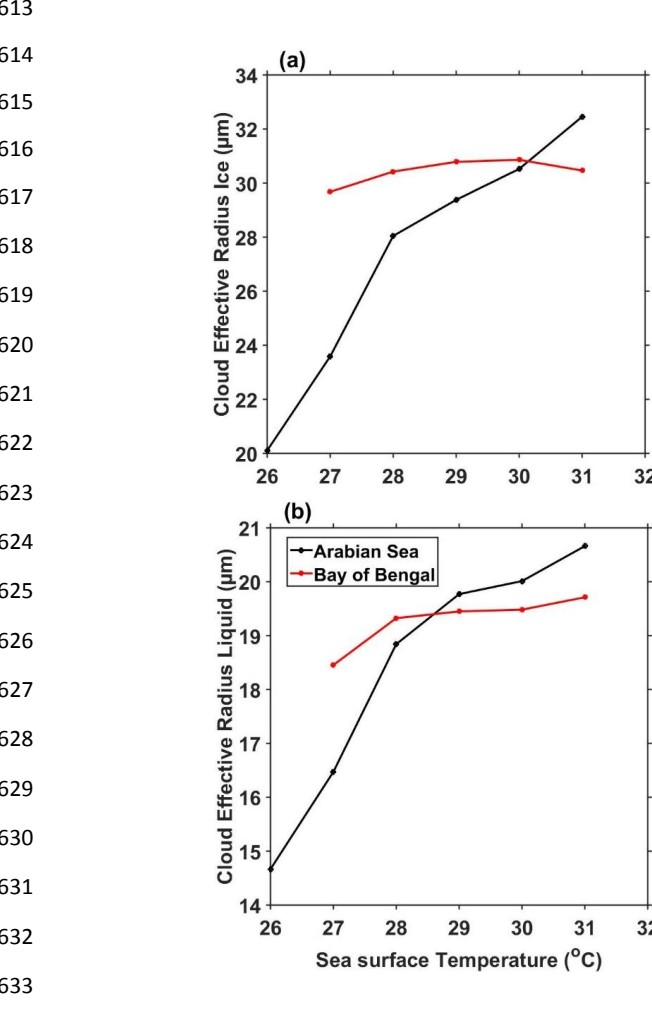

**Figure 5:** (a) and (b), respectively, represent the variation of mean CER ice (in μm) and mean CER liquid (in μm) with SST over the AS (63°E-72°E & 8°N-20°N) and the BOB (83°E-92°E & 8°N-21°N) during the SWM season.






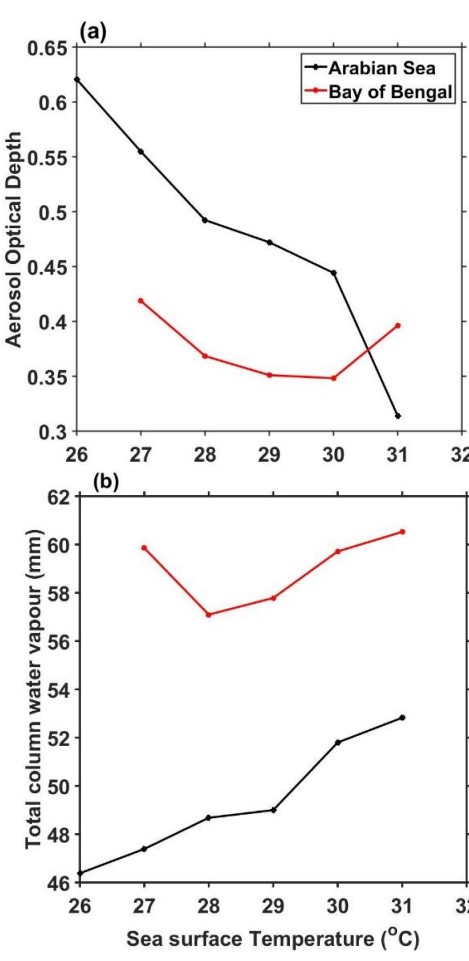

**Figure 6:** (a) The variation of mean AOD and (b) TCWV (in mm) with SST over the AS
(63°E-72°E & 8°N-20°N) and the BOB (83°E-92°E & 8°N-21°N) during SWM.















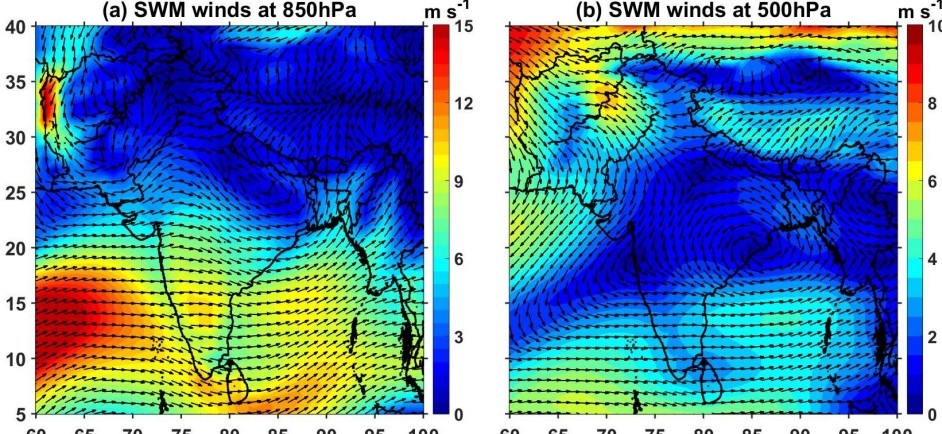


**Figure 7:** Winds during the SWM season at 850 hPa and 500 hPa levels. The shading colors represent the magnitude of the wind and arrow indicates the direction of the wind.






















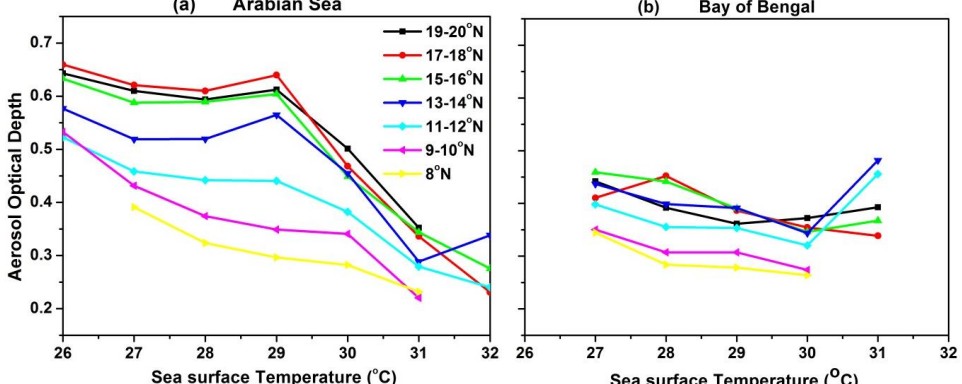


**Figure 8:** (a) and (b), respectively, represent latitudinal variation (for every $2^o$ latitude interval) of mean AOD over the AS (between 63°E and $72^o$E) and the BOB (between 83°E and 92°E).