# Peer review of "Variability of vertical structure of precipitation with sea surface temperature over the Arabian Sea and the Bay of Bengal as inferred by TRMM PR measurements"

_Atmospheric Chemistry and Physics, 2018_

## Referee Comment (RC1) · Anonymous Referee #2 · 2 Oct 2018

Please find the comments and suggestions on the manuscript (acp-2018-638 ) "Variability of vertical structure of precipitation with sea surface temperature over the Arabian Sea and the Bay of Bengal as inferred by TRMM PR measurements" by Kadiri Saikranthi, Basivi Radhakrishna, Thota Narayana Rao, and Sreedharan Krishnakumari Satheesh. First of all, I should complement the author for the good quality and novel work.

This manuscript is nicely crafted, starting from the defining the problem to the results. The results presented in this work address the key point on the vertical structure of

rain over Arabian sea (AS) and Bay of Bengal (BoB) also they related with the SST variations. This article is worthy to publish such kind of results in scientific journals like Atmos. Chem. Phys and I recommend for publication. But before publishing, I have made specific comments and recommend the authors to answer them before publishing.

Specific comments:

Figure 5: Why CER of the ice show a decreasing trend and CER of water showing an increasing trend over boB beyond 30°C? Whereas over AS, both CER liquid and Ice shows an increasing trend?

Figure 5: Why CER of ice(water) shows a reverse trend beyond 30°C(28.5°C) over AS and BoB.

Figures 2 and 5: Higher values of reflectivities beyond 8 km beyond 30°C over AS is due to the higher values of CER liquid (Fig. 5)? That means higher convection over AS than BoB?

Whether similar explanation holds good for LTS over AS?

Looking forward to the replies.

---

## Short Comment (SC1) · 13 Nov 2018

The article titled "Variability of vertical structure of precipitation with sea surface temperature over the Arabian Sea and the Bay of Bengal as inferred by TRMM PR measurements" submitted by Kadiri Saikranthi et al has to address about the following concerns before the Editor makes a final decision.

(a) The article title highlights aspect of the variability of vertical structure of precipitation with sea surface temperature (SST). However, the authors explore the relationships be-

tween the SST and other variables such as AOD, CER ice and CER liquid, total column water vapour etc. that may not directly represent the vertical structure of precipitation.

(b) The figure 1 shows the regions considered in this study with background colour representing the mean SST during SWM period over AS and BoB. It is clearly evident that the regions of interest depict significant spatial heterogeneity in the SST ($\sim$ 2 degrees C). In such a scenario, (in the figures 4, 5 and 6) I think the standard deviation should be present in those figures.

(c) I would recommend to use MODIS level 2 data products for AOD, CER-ice and CER-liquid for exploring the relationships between different variables. Further, the authors have not mentioned from where the total column water vapour data was obtained. Even the combined uncertainty from different sources of data (e.g., TRMM, MODIS and ECMWF Interim Reanalysis) was not accounted for when establishing the relationships.

(d) It would be nice if the authors establish the mechanism on why the contrasting relationships were observed over BoB and AS. The authors shall note that SST depends on other factors such as turbidity of the sea water and sea surface albedo, which in turn depend on other variables including wind speed and chlorophyll concentration. While the authors have ignored these essential variables, the relationships with AOD, CER-ice, CER-liquid and total column water vapour alone cannot provide the variability in SST in the regions of interest.

---

## Referee Comment (RC2) · Anonymous Referee #4 · 4 Dec 2018

Title: Variability of the vertical structure of precipitation with sea surface temperature over the Arabian Sea and Bay of Bengal as inferred by TRMM PR measurements Authors: Saikranthi et al. Recommendation: Rejection

Scientific significance: Fair Scientific Quality: Poor Presentation Quality: Poor

This study investigated the variability in the vertical structure of precipitation as a function of sea surface temperature using TRMM precipitation radar measurements. I think the paper lacks focus, inadequate analysis, and insufficient literature review. The intent of the paper digresses at some point by incorporating the aerosol/cloud radiation analysis without a context jumbling both convective dynamics and radiative impacts of aerosols on clouds. Given the scope, the section with aerosol and radiation properties are redundant. Most of the analysis lacks context. Overall, the quality and the content of the present paper is poor.

Comments on Introduction: Introduction lacks discussion on how Arabian sea and Bay of Bengal regions are distinctly different in its background state, which would help them explain the further analysis on convective profiles. Though the authors have claimed to have studied the "causative mechanisms" of SST with the vertical structure of precipitation in the introduction, no suggestions based on the analysis performed have been discussed in the later sections. Mere correlation doesn't explain the causality, which needs carefully controlled model experiments with a rigor to assess the confounding factors controlling the SST and precipitation relationship.

Comments on the analysis: Given the non-linear influence of sea surface temperature on the variability of precipitation structure, it would be an oversimplification to look at the influence of SST on the mean structure of radar echoes. It would have been interesting to classify the mean structure further into different cloud types (e.g., shallow/congestus/deep/) and assess the variability of these populations in terms of factors (e.g., winds, stability) that are co-associated with SSTs. There are no insights been provided on why the differences in the variabilities of vertical structure exist between AS and BOB. It is important to investigate if more variability over the AS is due to fluctuations in the winds/SSTs or both. From figure 2, it is evident that AS region has more seasonality in term of air-sea variables compared to BOB. Given the influence of more variables, merely analyzing indirect relationships of precipitation structure with SSTs would be futile. One way to analyze is to look at the variability of large-scale parameters (e.g., stability, vertical velocity, wind speed) for a given SST, and look at the cloud population in terms of these co-associated variables. By doing so, one would prioritize the combination of factors that lead to different convection type. SST influence on the

clouds is of the first order, however, it is also important to show the temporal variation, highlighting the seasonal evolution of cloud types collocated with SSTs and other variables.

In its entirety, the quality of the scientific analysis is insufficient, lacking depth and focus. At this moment, I don't see the presentable quality and content of the paper is ready for publishing. Hence, I have to reject the paper in its present format. I hope the authors would consider revising the paper with the proposed suggestions and resubmit the paper once its ready.

Specific comments:

1. The stability measure (LTS) used here is appropriate for stratiform clouds, which may not be appropriate for convective clouds in these regions. One may use static stability profiles instead.

---

## Author Comment (AC1) · 2 Jan 2019

Replies to reviewer #2

At the outset, we thank the reviewer for positive and constructive comments that improved the quality of the manuscript.

Comment: Figure 5: Why CER of the ice show a decreasing trend and CER of water showing an increasing trend over BOB beyond 30°C? Whereas over AS, both CER liquid and Ice shows an increasing trend?

Reply: The main reason for studying CER at different SST is to understand whether or not the observed differences originated at the formation of cloud stage. For that, CER for water is sufficient. Therefore, figure and text related to CER for ice are removed from the revised manuscript. Regarding reviewers' query, yes, there are some small differences in the variation of CER for ice and water with SST above 30 °C, but they are not significant.

Comment: Figure 5: Why CER of ice (water) shows a reverse trend beyond 30°C (28.5°C) over AS and BoB. Reply: The CER depends on the ambient atmospheric aerosol concentration and availability of water vapor. The variation of AOD with SST is substantial over the AS while it is marginal over the BOB. As the SST increases AOD decreases and TCWV increases results in increase in CER over the AS and is more prominent at higher SSTs (where the decrease of AOD with SST is quite substantial). On the other hand, the decrease in AOD with SST is quite marginal over BOB and in fact, AOD increases from 30 °C to 31 °C. Therefore, the CER for water continuously increases with rapid increase beyond 28 °C over AS, while the increase is marginal over BOB.

Comment: Figures 2 and 5: Higher values of reflectivities beyond 8 km beyond 30°C over AS is due to the higher values of CER liquid (Fig. 5)? That means higher convection over AS than BOB? Whether similar explanation holds good for LTS over AS? Reply: The differences in Z over AS and BOB at and above 8 km is very small (within 1 dBZ) and not significant. Therefore, we are not attributing these to any physical or microphysical processes.

---

## Author Comment (AC2) · 2 Jan 2019

*At the outset thank Mr. B. Guha for reading our manuscript and suggesting comments.*

**Comment:** (a) The article title highlights aspect of the variability of vertical structure of precipitation with sea surface temperature (SST). However, the authors explore the relationships between the SST and other variables such as AOD, CER ice and CER liquid, total column water vapour etc. that may not directly represent the vertical structure of precipitation.

*Reply: The generation and growth of clouds and precipitating systems depend on the triggering mechanisms (over Oceans, it is primarily SST) and ambient dynamical and thermodynamical environment (Houze et al., 2015). Changes in SST have the potential of altering the type of precipitating system and the vertical structure of precipitation (Oueslati and Bellon 2015). Besides the SST, vertical structure can be modified by aerosols (or CCN, mostly at the cloud formation stage) and thermodynamics of the ambient atmosphere. For instance, recent studies have shown the impact of surface aerosols ($PM_{10}$) in altering the vertical structure of precipitation (Gao et al., 2018 and references therein). We, therefore, need to understand the observed variations exist at the cloud formation stage or manifested during the descent of precipitation particles to the ground. The cloud effective radius (CER for water) (depend on aerosols and TCWV) is a good proxy to understand the cloud microphysical processes. While, vertical velocity, winds, stability parameters are considered to depict the ambient atmosphere, which can alter the vertical structure of precipitation. All these parameters are considered in the present study to understand the vertical structure of precipitation over AS and BOB.*

**Comment:** (b) The figure 1 shows the regions considered in this study with background colour representing the mean SST during SWM period over AS and BOB. It is clearly evident that the regions of interest depict significant spatial heterogeneity in the SST (_ 2 degrees C). In such a scenario, (in the figures 4, 5 and 6) I think the standard deviation should be present in those figures.

*Reply: We wish to inform the reviewer that the segregation of SST data into different bins (26° to 31℃ with 1 interval) is done not by averaging the spatial data, rather using 1° X 1° gridded data. Therefore, there is no need to average the SST data. Instead, we provided standard deviation/standard error of mean values for CER, AOD, TVWV and vertical profiles of Z in the revised manuscript.*

**Comment:** (c) I would recommend to use MODIS level 2 data products for AOD, CER-ice and CER-liquid for exploring the relationships between different variables. Further, the authors have not mentioned from where the total column water vapour data was obtained. Even the combined uncertainty from different sources of data (e.g., TRMM, MODIS and ECMWF Interim Reanalysis) was not accounted for when establishing the relationships.

*Reply: The total column water vapor data are taken from the ERA-Interim reanalysis and this information is included in the revised manuscript. The spatial resolutions of MODIS level-2*

*and ERA-Interim SST are different. Thus, to know the values of AOD and CER at different SSTs, again the MODIS level-2 dataset needs to be regridded. Instead of regridding, we have used equal spatial lengths MODIS level-3 and SST datasets.*

**Comment:** (d) It would be nice if the authors establish the mechanism on why the contrasting relationships were observed over BOB and AS. The authors shall note that SST depends on other factors such as turbidity of the sea water and sea surface albedo, which in turn depend on other variables including wind speed and chlorophyll concentration. While the authors have ignored these essential variables, the relationships with AOD, CER ice, CER-liquid and total column water vapour alone cannot provide the variability in SST in the regions of interest.

*Reply: We do agree that SST over open Oceans depends on many factors. But our interest is not to show how precipitating systems alter the SST over the AS and BOB. Rather, we focused on the variation of vertical structure of precipitation (in terms of precipitation top height and intensity) with SST over the AS and the BOB and the factors responsible for the variations in the vertical structure over both these oceans.*

---

## Author Comment (AC3) · 2 Jan 2019

Referee #4

At the outset, we thank the reviewer for positive and constructive comments that improved the quality of the manuscript.

Comment: This study investigated the variability in the vertical structure of precipitation as a function of sea surface temperature using TRMM precipitation radar measurements. I think the paper lacks focus, inadequate analysis, and insufficient literature review. The intent of the paper digresses at some point by incorporating the aerosol/cloud radiation analysis without a context jumbling both convective dynamics and radiative impacts of aerosols on clouds. Given the scope, the section with aerosol and radiation properties is redundant. Most of the analysis lacks context. Overall, the quality and the content of the present paper are poor.

Reply: The aim of the present study is to understand differences in the variation of vertical structure of precipitation with SST over the Arabian Sea and Bay of Bengal. SST being the main driving force to trigger precipitating systems through air-sea interactions, the occurrence of precipitation top height and intensity profiles (reflectivity) as a function of SST are studied. Besides SST, the vertical structure can be modified by aerosols (or CCN, mostly at the cloud formation stage) and thermodynamics of the ambient atmosphere. In the revised manuscript, all these parameters are considered to explain the differences in the vertical structure. Aerosols are considered only for understanding variation in cloud effective radius, nevertheless their radiative effects (direct, indirect, etc.) are not considered in the present study. Recent studies, indeed, have shown the impact of aerosols (PM10) on the vertical structure of precipitation (Gao et al., 2018 and references therein). We have rewritten the introduction with more focus on the above aspects and highlighting the known differences in various aspects/parameters over AS and BOB. The literature survey is also improved considerably in the revised manuscript by adding appropriate references (Guo et al. 2018; Nuijens et al. 2017; Weller et al. 2016; Sathiyamoorthy et al. 2013; Takayabu et al. 2010; Bhat et al. 2001; Ramanathan et al. 2001; Gadgil 2000; Krishnamurti 1981; Narayanan and Rao 1981;).

Comment: Introduction lacks discussion on how Arabian Sea and Bay of Bengal regions are distinctly different in its background state, which would help them explain the further analysis on convective profiles. Though the authors have claimed to have studied the "causative mechanisms" of SST with the vertical structure of precipitation in the introduction, no suggestions based on the analysis performed have been discussed in

the later sections. Mere correlation doesn't explain the causality, which needs carefully controlled model experiments with a rigor to assess the confounding factors controlling the SST and precipitation relationship.

Reply: The introduction of the revised manuscript is modified by considering all the suggestion of the reviewer. The role of the surrounding seas on the rainfall over the Indian landmass is stated and the differences between the two seas are clearly mentioned with proper references in the revised manuscript as follows:

[revised manuscript text omitted]

Comment: Given the non-linear influence of sea surface temperature on the variability of precipitation structure, it would be an oversimplification to look at the influence of SST on the mean structure of radar echoes. It would have been interesting to classify the mean structure further into different cloud types (e.g., shallow/congestus/deep/) and assess the variability of these populations in terms of factors (e.g., winds, stability) that are co-associated with SSTs. There are no insights been provided on why the differences in the variabilities of vertical structure exist between AS and BOB. It is important to investigate if more variability over the AS is due to fluctuations in the winds/SSTs or both. From figure 2, it is evident that AS region has more seasonality in term of air-sea variables compared to BOB. Given the influence of more variables, merely analyzing indirect relationships of precipitation structure with SSTs would be futile. One way to analyze is to look at the variability of large-scale parameters (e.g., stability, vertical velocity, wind speed) for a given SST, and look at the cloud population in terms of these co-associated variables. By doing so, one would prioritize the combination of factors that lead to different convection type. SST influence on the clouds is of the first order; however, it is also important to show the temporal variation, highlighting the seasonal evolution of cloud types collocated with SSTs and other variables.

Reply: We agree with the reviewer that all the forcing/controlling parameters (SST, winds, vertical wind velocity, stability, etc.) need to be considered for understanding the vertical structure of precipitation. We did the same in the revised version of the manuscript. Also, we studied the vertical structure of two types of precipitation (deep and shallow) as suggested by the reviewer. Since, stratiform rain is the trailing portions of convective complexes (Houze et al. 2015) and is not directly driven by the SST, it's relation with SST is not dealt separately.

Comment: The stability measure (LTS) used here is appropriate for stratiform clouds, which may not be appropriate for convective clouds in these regions. One may use

static stability profiles instead.

Reply: As suggested by the reviewer instead of LTS the static stability (profiles of equivalent potential temperature) is used in the revised manuscript to explain the convective strength as a function of SST.

Please also note the supplement to this comment:
https://www.atmos-chem-phys-discuss.net/acp-2018-638/acp-2018-638-AC3-supplement.pdf
* * *

---

## Author Response (AR4)

**Replies to reviewer #2**

*At the outset, we thank the reviewer for positive and constructive comments that improved the quality of the manuscript.*

**Comment:** Figure 5: Why CER of the ice show a decreasing trend and CER of water showing an increasing trend over BOB beyond 30°C? Whereas over AS, both CER liquid and Ice shows an increasing trend?

*Reply: The main reason for studying CER at different SST is to understand whether or not the observed differences originated at the formation of cloud stage. For that, CER for water is sufficient. Therefore, figure and text related to CER for ice are removed from the revised manuscript.*

*Regarding reviewers' query, yes, there are some small differences in the variation of CER for ice and water with SST above 30 °C, but they are not significant.*

**Comment:** Figure 5: Why CER of ice (water) shows a reverse trend beyond 30°C (28.5°C) over AS and BoB.

*Reply: The CER depends on the ambient atmospheric aerosol concentration and availability of water vapor. The variation of AOD with SST is substantial over the AS while it is marginal over the BOB. As the SST increases AOD decreases and TCWV increases results in increase in CER over the AS and is more prominent at higher SSTs (where the decrease of AOD with SST is quite substantial). On the other hand, the decrease in AOD with SST is quite marginal over BOB and in fact, AOD increases from 30 °C to 31 °C. Therefore, the CER for water continuously increases with rapid increase beyond 28 °C over AS, while the increase is marginal over BOB.*

**Comment:** Figures 2 and 5: Higher values of reflectivities beyond 8 km beyond 30°C over AS is due to the higher values of CER liquid (Fig. 5)? That means higher convection over AS than BOB?

Whether similar explanation holds good for LTS over AS?

*Reply: The differences in Z over AS and BOB at and above 8 km is very small (within 1 dBZ) and not significant. Therefore, we are not attributing these to any physical or microphysical processes.*

*At the outset, we thank the reviewer for positive and constructive comments that improved the quality of the manuscript.*

**Comment:** This study investigated the variability in the vertical structure of precipitation as a function of sea surface temperature using TRMM precipitation radar measurements. I think the paper lacks focus, inadequate analysis, and insufficient literature review. The intent of the paper digresses at some point by incorporating the aerosol/cloud radiation analysis without a context jumbling both convective dynamics and radiative impacts of aerosols on clouds. Given the scope, the section with aerosol and radiation properties is redundant. Most of the analysis lacks context. Overall, the quality and the content of the present paper are poor.

*Reply: The aim of the present study is to understand differences in the variation of vertical structure of precipitation with SST over the Arabian Sea and Bay of Bengal. SST being the main driving force to trigger precipitating systems through air-sea interactions, the occurrence of precipitation top height and intensity profiles (reflectivity) as a function of SST are studied. Besides SST, the vertical structure can be modified by aerosols (or CCN, mostly at the cloud formation stage) and thermodynamics of the ambient atmosphere. In the revised manuscript, all these parameters are considered to explain the differences in the vertical structure. Aerosols are considered only for understanding variation in cloud effective radius, nevertheless their radiative effects (direct, indirect, etc.) are not considered in the present study. Recent studies, indeed, have shown the impact of aerosols ($PM_{10}$) on the vertical structure of precipitation (Gao et al., 2018 and references therein).*

*We have rewritten the introduction with more focus on the above aspects and highlighting the known differences in various aspects/parameters over AS and BOB. The literature survey is also improved considerably in the revised manuscript by adding appropriate references (Guo et al. 2018; Nuijens et al. 2017; Weller et al. 2016; Sathiyamoorthy et al. 2013; Takayabu et al. 2010; Bhat et al. 2001; Ramanathan et al. 2001; Gadgil 2000; Krishnamurti 1981; Narayanan and Rao 1981;).*

**Comment:** Introduction lacks discussion on how Arabian Sea and Bay of Bengal regions are distinctly different in its background state, which would help them explain the further analysis on convective profiles. Though the authors have claimed to have studied the "causative mechanisms" of SST with the vertical structure of precipitation in the introduction, no suggestions based on the analysis performed have been discussed in the later sections. Mere correlation doesn't explain the causality, which needs carefully controlled model experiments with a rigor to assess the confounding factors controlling the SST and precipitation relationship.

*Reply: The introduction of the revised manuscript is modified by considering all the suggestion of the reviewer. The role of the surrounding seas on the rainfall over the Indian landmass is stated and the differences between the two seas are clearly mentioned with proper references in the revised manuscript as follows:*

[revised manuscript text omitted]

**Comment:** Given the non-linear influence of sea surface temperature on the variability of precipitation structure, it would be an oversimplification to look at the influence of SST on the mean structure of radar echoes. It would have been interesting to classify the mean structure further into different cloud types (e.g., shallow/congestus/deep/) and assess the variability of these populations in terms of factors (e.g., winds, stability) that are co-associated with SSTs. There are no insights been provided on why the differences in the variabilities of vertical structure exist between AS and BOB. It is important to investigate if more variability over the AS is due to fluctuations in the winds/SSTs or both. From figure 2, it is evident that AS region has more seasonality in term of air-sea variables compared to BOB. Given the influence of more variables, merely analyzing indirect relationships of precipitation structure with SSTs would be futile. One way to analyze is to look at the variability of large-scale parameters (e.g., stability, vertical velocity, wind speed) for a given SST, and look at the cloud population in terms of these co-associated variables. By doing so, one would prioritize the combination of factors that lead to different convection type. SST influence on the clouds is of the first order; however, it is also important to show the temporal variation, highlighting the seasonal evolution of cloud types collocated with SSTs and other variables.

*Reply: We agree with the reviewer that all the forcing/controlling parameters (SST, winds, vertical wind velocity, stability, etc.) need to be considered for understanding the vertical structure of precipitation. We did the same in the revised version of the manuscript. Also, we studied the vertical structure of two types of precipitation (deep and shallow) as suggested by the reviewer. Since, stratiform rain is the trailing portions of convective complexes (Houze et al. 2015) and is not directly driven by the SST, it's relation with SST is not dealt separately.*

**Comment:** The stability measure (LTS) used here is appropriate for stratiform clouds, which may not be appropriate for convective clouds in these regions. One may use static stability profiles instead.

*Reply: As suggested by the reviewer instead of LTS the static stability (profiles of $\theta_e$) is used in the revised manuscript to explain the convective strength as a function of SST.*

**Replies to short comments**

*At the outset thank Mr. B. Guha for reading our manuscript and suggesting comments.*

**Comment:** (a) The article title highlights aspect of the variability of vertical structure of
precipitation with sea surface temperature (SST). However, the authors explore the
relationships between the SST and other variables such as AOD, CER ice and CER liquid,
total column water vapour etc. that may not directly represent the vertical structure of
precipitation.

*Reply: The generation and growth of clouds and precipitating systems depend on the*
*triggering mechanisms (over Oceans, it is primarily SST) and ambient dynamical and*
*thermodynamical environment (Houze et al., 2015). Changes in SST have the potential of*
*altering the type of precipitating system and the vertical structure of precipitation (Oueslati*
*and Bellon 2015). Besides the SST, vertical structure can be modified by aerosols (or CCN,*
*mostly at the cloud formation stage) and thermodynamics of the ambient atmosphere. For*
*instance, recent studies have shown the impact of surface aerosols ($PM_{10}$) in altering the*
*vertical structure of precipitation (Gao et al., 2018 and references therein). We, therefore,*
*need to understand the observed variations exist at the cloud formation stage or manifested*
*during the descent of precipitation particles to the ground. The cloud effective radius (CER*
*for water) (depend on aerosols and TCWV) is a good proxy to understand the cloud*
*microphysical processes. While, vertical velocity, winds, stability parameters are considered*
*to depict the ambient atmosphere, which can alter the vertical structure of precipitation. All*
*these parameters are considered in the present study to understand the vertical structure of*
*precipitation over AS and BOB.*

**Comment:** (b) The figure 1 shows the regions considered in this study with background
colour representing the mean SST during SWM period over AS and BOB. It is clearly
evident that the regions of interest depict significant spatial heterogeneity in the SST (_ 2
degrees C). In such a scenario, (in the figures 4, 5 and 6) I think the standard deviation should
be present in those figures.

*Reply: We wish to inform the reviewer that the segregation of SST data into different bins*
*(26° to 31°C with 1 interval) is done not by averaging the spatial data, rather using 1° X 1°*
*gridded data. Therefore, there is no need to average the SST data. Instead, we provided*
*standard deviation/standard error of mean values for CER, AOD, TVWV and vertical profiles*
*of Z in the revised manuscript.*

**Comment:** (c) I would recommend to use MODIS level 2 data products for AOD, CER-ice
and CER-liquid for exploring the relationships between different variables. Further, the
authors have not mentioned from where the total column water vapour data was obtained.
Even the combined uncertainty from different sources of data (e.g., TRMM, MODIS and
ECMWF Interim Reanalysis) was not accounted for when establishing the relationships.

*Reply: The total column water vapor data are taken from the ERA-Interim reanalysis and this*
*information is included in the revised manuscript. The spatial resolutions of MODIS level-2*

*and ERA-Interim SST are different. Thus, to know the values of AOD and CER at different*
*SSTs, again the MODIS level-2 dataset needs to be regridded. Instead of regridding, we have*
*used equal spatial lengths MODIS level-3 and SST datasets.*

**Comment:** (d) It would be nice if the authors establish the mechanism on why the
contrasting relationships were observed over BOB and AS. The authors shall note that SST
depends on other factors such as turbidity of the sea water and sea surface albedo, which in
turn depend on other variables including wind speed and chlorophyll concentration. While
the authors have ignored these essential variables, the relationships with AOD, CER ice,
CER-liquid and total column water vapour alone cannot provide the variability in SST in the
regions of interest.

***Reply:*** *We do agree that SST over open Oceans depends on many factors. But our interest is*
*not to show how precipitating systems alter the SST over the AS and BOB. Rather, we focused*

[revised manuscript text omitted]

**Replies to Reviewer**

*At the outset, we thank the reviewer for positive and constructive comments that improved the quality of the manuscript.*

**Comment:** It is not clear the SST effect is the primary cause of the variability of vertical structure of precipitation. The authors should examine the SST effect on this variability under similar monsoon westerlies conditions, following the paper by Takahashi and Dado (2018) showing that SST makes a positive contribution toward rainfall in the Philippines during the summer Monsoon, but the monsoon westerly is the primary driver of the variation in rainfall.

**Reply:** *It is true that SST alone cannot explain all the observed variability. SST, of course, is the main forcing parameter, but the vertical structure is dictated by several atmospheric factors, like temperature inversions, atmospheric instability, availability of moisture (in the mid-troposphere), wind shear, etc. Takahashi and Dado (2018) have shown that zonal wind variations can also explain some variability of rain. To examine the impact of zonal wind on rainfall over the Arabian Sea and Bay of Bengal, the data are segregated into 3 wind regimes as weak (monsoon westerlies lies between 0 and 6 m s$^{-1}$), moderate (monsoon westerlies lies between 6 to 12 m s$^{-1}$) and strong (monsoon westerlies > 12 m s$^{-1}$) winds. The median vertical profiles of reflectivity are computed for each SST bin for deep and shallow systems. These median reflectivity profiles for shallow and deep systems at each SST bin and for each wind category are shown in Figures R1, R2, R3 & R4. Two important observations are noted from these figures. 1. Vertical profiles of reflectivity show considerable variation (2-5 dBZ) in all wind categories over the Arabian Sea, but such variations are absent over the Bay of Bengal. It implies that the reported differences in reflectivity profiles over the Arabian Sea and Bay of Bengal exist in all wind regimes. 2. The variation in reflectivity with SST increases with weak to strong wind regime over the Arabian Sea, indicating some influence of wind on reflectivity (rainfall) variation.*

*The above information is included in the revised manuscript (but not figures).*

[Figure]

**Figure R1:** *Vertical profiles of median reflectivity and standard deviation during weak, moderate and strong westerly wind regimes corresponding to deep systems as a function of SST over the AS during the ISM season. Also shown are the number of conditional reflectivity pixels at each altitude used for the estimation of the median and standard deviation.*

[Figure]

**Figure R2:** *Vertical profiles of median reflectivity and standard deviation during weak, moderate and strong westerly wind regimes corresponding to deep systems as a function of SST over the BOB during the ISM season. Also shown are the number of conditional reflectivity pixels at each altitude used for the estimation of the median and standard deviation.*

[Figure]

**Figure R3:** *Vertical profiles of median reflectivity and standard deviation during weak, moderate and strong westerly wind regimes corresponding to shallow systems as a function of SST over the AS during the ISM season. Also shown are the number of conditional reflectivity pixels at each altitude used for the estimation of the median and standard deviation.*

[Figure]

**Figure R4:** *Vertical profiles of median reflectivity and standard deviation during weak, moderate and strong westerly wind regimes corresponding to shallow systems as a function of SST over the BOB during the ISM season. Also shown are the number of conditional reflectivity pixels at each altitude used for the estimation of the median and standard deviation.*

**Comment:** I cannot agree with the arguments described in Line 283-285, since cloud effective radius (CER) is not simply linked to precipitation size (i.e. Z-R). The authors should refer to the review by Rosenfeld, D., and C. W. Ulbrich (2003) describing that microphysically "continental" clouds with greater concentrations of small cloud droplets produce greater concentrations of large rain drops and smaller concentrations of small rain drops compared to microphysically "maritime" clouds with small concentrations of large cloud droplets.

**Reply:** *We do agree with the reviewer that is it not entirely correct to directly link CER to raindrop size (reflectivity), because several microphysical and dynamical processes occur during the cloud drop growth (collision-coalescence, riming, etc.) to rain drop and also its descent (evaporation, etc.) to the ground (Rosenfeld and Ulbrich, 2003; Rao et al., 2009; Radhakrishna et al. 2009). The slope of the vertical profile of reflectivity can provide the dominant microphysical processes occurring during the drop evolution (Saikranthi et al. 2014; Rao et al. 2016). In the present study, the reflectivity gradients are negative (i.e., reflectivity increases in magnitude with decreasing height) at all SST's, albeit with varying magnitude. It indicates that, on average, there is a low-level hydrometeor growth at all SST's over both Arabian Sea and Bay of Bengal. Since the microphysical process at all SST's is same, we linked CER and raindrop size. Since both the study regions (Arabian Sea and Bay of Bengal) are oceanic regions, it is a reasonable approximation. However, such approximations may not necessarily be valid over continental regions (or continental clouds) and dry regions (where evaporation of raindrops plays a dominant role) (Radhakrishna et al. 2009; Saikranthi et al. 2014; Rao et al. 2006).*

**Comment:** It is not clear the definition of deep systems shown in Fig. 3. If deep systems include both convection and stratiform precipitation, they should be separated. I would suggest the authors refer to the paper by Kobayashi et al. (2018) describing vertical gradient of stratiform radar reflectivity below the bright band.

**Reply:** *The main objective of the paper is to understand the impact of SST (and other atmospheric processes) on the vertical structure of precipitation. Since SST is the surface forcing parameter and triggers only convection (could be shallow or deep)(here convection means the physical process not the type of rain, which we generally refer to as convective rain)(Houze et al. 2015), we primarily focused on these two types of systems. Stratiform rain is the trailing or decaying portion of the convective cell, so, we may not find a direct link between SST and stratiform rain.*

**Comment:** I speculate that small variation of vertical structure of precipitation with SST over BoB should be explained by the fact that rainfall over BoB is produced by southeastward-propagating systems from the India coast (Yang and Slingo 2001; Li and Carbone 2015) rather than those developed in situ.

**Reply:** *We do agree with the reviewer that the systems generating along the east coast of India propagate towards the Bay of Bengal at diurnal scale. The local conditions (including SST) play an important role for the propagation of systems (they are not simple advective systems). In that context, it is important to check all the background parameters. All these parameters, like vertical velocity, horizontal wind gradients, AOD, CER and columnar water*

*vapor, show smaller variation with SST over the Bay of Bengal than Arabian Sea, indicating that atmospheric conditions are entirely different over the Arabian Sea and Bay of Bengal and are dictating the vertical structure of precipitation.*

**Comment:** Line 78-80: I would suggest that the authors refer to Kumar et al. (2014) and Shige et al. (2017) describing summer monsoon rainfall over the Western Ghats and Myanmar coast.

**Reply:** *The above references are added in the introduction of the revised manuscript.*

[revised manuscript text omitted]

*At the outset we thank the reviewer for constructive comments that improved the quality of the manuscript.*

**Comment**: Under weak wind regimes, the reflectivity for deep and shallow systems decreases with increase in SST over AS (Figs. R1 and R3). Why? Is this consistent with the authors' argument that the variations seen in reflectivity are originated in the cloud formations stage itself? This point should be discussed in the manuscript.

**Reply**: It is true that the reflectivity pattern with SST is somewhat different for weak wind category for reasons not known at present. But, it appears to be interesting and will be pursued later. However, our argument that the variations in reflectivity are originated in the cloud formation is based on TCW, CER and AOD data. To examine the validity of this statement at weak wind regime, we have segregated the above data for weak wind conditions and plotted below for reviewers' reference. The variation of TCW, CER and AOD with SST for weak wind regime is very similar to that of total data, indicating that our argument is still valid even for weak wind regime.

[Figure]

**Figure R1:** *Variation of mean and standard error of TCW (in kg m$^{-2}$), CER liquid (in μm) and AOD with SST over the AS during week wind regime.*

**Comment**: The authors should note their assumption that the microphysical process at all SST's is same in the manuscript.

**Reply**: Reflectivity profiles show an increase with decreasing height over both seas, albeit with varying magnitude. It indicates that the microphysical growth processes could be the same at all SST's but their efficacy could be different owing to ambient atmospheric conditions.

**Comment**: Again, the authors should explain the definition of deep and shallow systems (not just refer to Houze et al. 2007) in the manuscript.

**Reply**: *The deep and shallow systems definitions are included in the revised manuscript. "Profiles are classified as deep (shallow), if their storm top reflectivity ≥ 17 dBZ lies above (1 km below) the 0°C isotherm".*

**Comment**: L38: Tropical rainfall measuring mission should be Tropical Rainfall Measuring Mission.

**Reply**: *The typos are corrected in the revised manuscript.*

**Comment**: L92: tropical rainfall measuring mission should be Tropical Rainfall Measuring Mission.

**Reply**: *The typos are corrected in the revised manuscript.*

**Comment**: L103: I would suggest the authors add Tao et al. (2016).

**Reply**: *The reference is added in the revised manuscript.*

*Tao, W.-K., Y. N. Takayabu, S. Lang, S. Shige, W. Olson, A. Hou, G. Skofronick-Jackson, X. Jiang, C. Zhang, W. Lau, T. Krishnamurti, D. Waliser, M. Grecu, P. E. Ciesielski, R. H. Johnson, R. Houze, R. Kakar, N. Nakamura, S. Braun, R. Oki, and A. Bhardwaj, 2016: TRMM Latent Heating Retrieval: Applications and Comparisons with Field Campaigns and Large-Scale Analyses, Meteorological Monographs - Multi-scale Convection-Coupled Systems in the Tropics: A tribute to Dr. Michio Yanai, 56, 2.1-2.34, DOI: 10.1175/AMSMONOGRAPHS-D-15-0013.1.*

**Comment**: L295: the particle size should be the precipitating particle size.

**Reply**: *'particle size' is replaced with 'precipitating particle size' in the revised manuscript.*

**Comment**: L305-307: Change in fall speed from ice hydrometers to raindrops should be added.

**Reply**: *The text "change in fall speed from ice hydrometers to raindrops" is added in the revised manuscript.*

**Comment**: p. 29: The figure should be Figure 7.  p. 30: The figure should be Figure 6.

**Reply**: *The figures are interchanged in the revised manuscript.*

[revised manuscript text omitted]